# ISS: Image as Stepping Stone for Text-Guided 3D Shape Generation

**Zhengzhe Liu**[1]  **Peng Dai**[2]  **Ruihui Li**[3]  **Xiaojuan Qi**[2*]  **Chi-Wing Fu**[1*]
[1]The Chinese University of Hong Kong  [2]The University of Hong Kong  [3]Hunan University

## ABSTRACT

Text-guided 3D shape generation remains challenging due to the absence of large paired text-shape dataset, the substantial semantic gap between these two modalities, and the structural complexity of 3D shapes. This paper presents a new framework called *Image as Stepping Stone* (ISS) for the task by introducing 2D image as a stepping stone to connect the two modalities and to eliminate the need for paired text-shape data. Our key contribution is a *two-stage feature-space-alignment approach* that maps CLIP features to shapes by harnessing a pre-trained single-view reconstruction (SVR) model with multi-view supervisions: first map the CLIP image feature to the detail-rich shape space in the SVR model, then map the CLIP text feature to the shape space and optimize the mapping by encouraging CLIP consistency between the input text and the rendered images. Further, we formulate a *text-guided shape stylization module* to dress up the output shapes with novel structures and textures. Beyond existing works on 3D shape generation from text, our new approach is general for creating shapes in a broad range of categories, *without* requiring paired text-shape data. Experimental results manifest that our approach outperforms the state-of-the-arts and our baselines in terms of *fidelity* and *consistency with text*. Further, our approach can stylize the generated shapes with both realistic and fantasy structures and textures. Codes are available at `https://github.com/liuzhengzhe/ISS-Image-as-Stepping-Stone-for-Text-Guided-3D-Shape-Generation`.

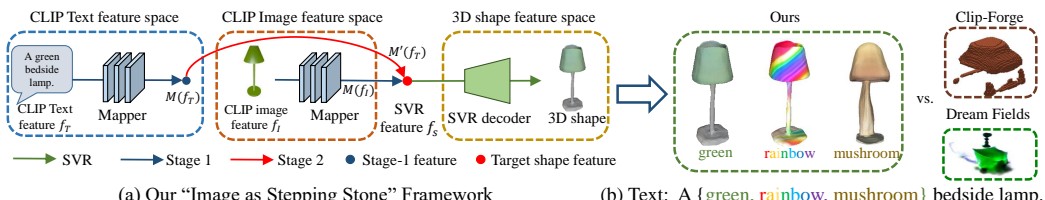

(a) Our "Image as Stepping Stone" Framework  (b) Text: A {green, rainbow, mushroom} bedside lamp.

Figure 1: Our novel "Image as Stepping Stone" framework (a) is able to connect the text space (the CLIP Text feature) and the 3D shape space (the SVR feature) through our two-stage feature-space alignment, such that we can generate plausible 3D shapes from text (b) beyond the capabilities of the existing works (CLIP-Forge and Dream Fields), without requiring paired text-shape data.

## 1 INTRODUCTION

3D shape generation has a broad range of applications, *e.g.*, in Metaverse, CAD, games, animations, etc. Among various ways to generate 3D shapes, a user-friendly approach is to generate shapes from natural language or text descriptions. By this means, users can readily create shapes, *e.g.*, to add/modify objects in VR/AR worlds, to design shapes for 3D printing, etc. Yet, generating shapes from texts is very challenging, due to the lack of large-scale paired text-shape data, the large semantic gap between the text and shape modalities, and the structural complexity of 3D shapes.

Existing works (Chen et al., 2018; Jahan et al., 2021; Liu et al., 2022) typically rely on paired text-shape data for model training. Yet, collecting 3D shapes is already very challenging on its own, let alone the tedious manual annotations needed to construct the text-shape pairs. To our best knowledge, the largest existing paired text-shape dataset (Chen et al., 2018) contains only two categories, *i.e.*, table and chair, thus severely limiting the applicability of the existing works.

Very recently, two annotation-free approaches, CLIP-Forge (Sanghi et al., 2022) and Dream Fields (Jain et al., 2022), were proposed to address the dataset limitation. These two state-of-the-art approaches attempt to utilize the joint text-image embedding from the large-scale pre-trained language vision model, *i.e.*, CLIP (Radford et al., 2021), to eliminate the need of requiring paired text-shape data in model training. However, it is still extremely challenging to generate 3D shapes from text without paired texts and shapes for the following reasons. First, the range of object categories that can be generated are still limited due to the scarcity of 3D datasets. For example, Clip-Forge (Sanghi et al., 2022) is built upon a shape auto-encoder; it can hardly generate plausible shapes beyond the ShapeNet categories. Also, it is challenging to learn 3D prior of the desired shape from texts. For instance, Dream Field (Jain et al., 2022) cannot generate 3D shapes like our approach due to the lack of 3D prior, as it is trained to produce only multi-view images with a neural radiance field. Further, with over an hour of optimization for each shape instance from scratch, there is still no guarantee that the multi-view consistency constraint of Dream Field (Jain et al., 2022) can enforce the model for producing shapes that match the given text; we will provide further investigation in our experiments. Last, the visual quality of the generated shapes is far from satisfactory due to the substantial semantic gap between the unpaired texts and shapes. As shown in Figure 1 (b), the results generated by Dream Field typically look surrealistic (rather than real), due to insufficient information extracted from text for the shape structures and details. On the other hand, CLIP-Forge (Sanghi et al., 2022) is highly restricted by the limited $64^3$ resolution and it lacks colors and textures, further manifesting the difficulty of generating 3D shapes from unpaired text-shape data.

Going beyond the existing works, we propose a new approach to 3D shape generation from text without needing paired text-shape data. Our key idea is to *implicitly leverage 2D image* as a stepping stone (ISS) to connect the text and shape modalities. Specifically, we employ the joint text-image embedding in CLIP and train a CLIP2Shape mapper to *map the CLIP image features to a pre-trained detail-rich 3D shape space with multi-view supervisions*; see Figure 1 (a): stage 1. Thanks to the joint text-image embedding from CLIP, our trained mapper is able to connect the CLIP text features with the shape space for text-guided 3D shape generation. Yet, due to the gap between the CLIP text and CLIP image features, the mapped text feature may not align well with the destination shape feature; see the empirical analysis in Section 3.2. Hence, we further fine-tune the mapper specific to each text input by *encouraging CLIP consistency* between the rendered images and the input text to enhance the consistency between the input text and the generated shape; see Figure 1 (a): stage 2.

Our new approach advances the frontier of 3D shape generation from text in the following aspects. First, by taking image as a stepping stone, we make the challenging text-guided 3D shape generation task more approachable and cast it as a single-view reconstruction (SVR) task. Having said that, we learn 3D shape priors from the adopted SVR model directly in the feature space. Second, benefiting from the learned 3D priors from the SVR model and the joint text-image embeddings, our approach can produce 3D shapes in only 85 seconds vs. 72 minutes of Dream Fields (Jain et al., 2022). More importantly, our approach is able to produce plausible 3D shapes, *not* multi-view images, beyond the generation capabilities of the state-of-the-art approaches; see Figure 1 (b).

With our two-stage feature-space alignment, we already can generate shapes with good fidelity from texts. To further enrich the generated shapes with vivid textures and structures beyond the generative space of the pre-trained SVR model, we additionally design a text-guided stylization module to generate novel textures and shapes by encouraging consistency between the rendered images and the text description of the target style. We then can effectively fuse with the two-stage feature-space alignment to enable the generation of both realistic and fantasy textures and also shapes beyond the generation capability of the SVR model; see Figure 1 (b) for examples. Furthermore, our approach is compatible with different SVR models (Niemeyer et al., 2020; Alwala et al., 2022). For example, we can adopt SS3D (Alwala et al., 2022) to generate shapes from single-view in-the-wild images to broaden the range of categorical 3D shapes that our approach can generate, going beyond Sanghi et al. (2022), which can only generate 13 categories of ShapeNet. Besides, our approach can also work with the very recent approach GET3D (Gao et al., 2022) to generate high-quality 3D shapes from text; see our results in Section 4.

## 2 RELATED WORKS

**Text-guided image generation.** Existing text-guided image generation approaches can be roughly cast into two branches: (i) direct image synthesis (Reed et al., 2016a;b; Zhang et al., 2017; 2018;

Xu et al., 2018; Li et al., 2019; 2020; Qiao et al., 2019; Wang et al., 2021) and (ii) image generation with a pre-trained GAN (Stap et al., 2020; Yuan & Peng, 2019; Souza et al., 2020; Wang et al., 2020; Rombach et al., 2020; Patashnik et al., 2021; Xia et al., 2021). Yet, the above works can only generate images for limited categories. To address this issue, some recent works explore zero-shot text-guided image generation (Ramesh et al., 2021; Ding et al., 2021; Nichol et al., 2021; Liu et al., 2021; Ramesh et al., 2022) to learn to produce images of any category. Recently, Zhou et al. (2022) and Wang et al. (2022b) leverage CLIP for text-free text-to-image generation. Text-guided shape generation is more challenging compared with text-to-image generation. First, it is far more labor-intensive and difficult to prepare a large amount of paired text-shape data than paired text-image data, which can be collected from the Internet on a large scale. Second, the text-to-shape task requires one to predict full 3D structures that should be plausible geometrically and consistently in all views, beyond the needs in single-view image generation. Third, 3D shapes may exhibit more complex spatial structures and topology, beyond regular grid-based 2D images.

**Text-guided 3D generation.**    To generate shapes from text, several works (Chen et al., 2018; Jahan et al., 2021; Liu et al., 2022) rely on paired text-shape data for training. To avoid paired text-shape data, two very recent works, CLIP-Forge (Sanghi et al., 2022) and Dream Fields (Jain et al., 2022), attempt to leverage the large-scale pre-trained vision-language model CLIP. Yet, they still suffer from various limitations, as discussed in the third paragraph of Section 1. Besides 3D shape generation, some recent works utilize CLIP to manipulate a shape or NeRF with text (Michel et al., 2022; Jetchev, 2021; Wang et al., 2022a) and to generate 3D avatars (Hong et al., 2022). In this work, we present a new framework for generating 3D shape from text without paired text-shape data by our novel two-stage feature-space alignment. Our experimental results demonstrate the superiority of this work beyond the existing ones in terms of fidelity and text-shape consistency.

**Single-view reconstruction.**    Another topic related to this work is single-view reconstruction (SVR). Recently, researchers have explored SVR with meshes (Agarwal & Gopi, 2020), voxels (Zubić & Liò, 2021), and 3D shapes (Niemeyer et al., 2020). Further, to extend SVR to in-the-wild categories, Alwala et al. (2022) propose SS3D to learn 3D shape reconstruction using single-view images in hundreds of categories. In our work, we propose to harness an SVR model to map images to shapes, such that we can take 2D image as a stepping stone for producing shapes from texts. Yet, we perform the mapping and feature alignment implicitly in the latent space rather than explicitly.

## 3    METHODOLOGY

### 3.1    OVERVIEW

This work aims to generate 3D shape $S$ from text $T$. Overall, our idea is to map the CLIP features to the shape space of a pre-trained SVR model, such that we can leverage the joint text-image embeddings from CLIP and also the 3D generation capability of the SVR model to enhance the generation of 3D shape from text. Hence, our method only needs to be trained with multi-view RGB or RGBD images and the associated camera poses without paired text-shape data. As Figure 2 shows, our framework includes (i) image encoder $E_S$, which maps input image $I$ to SVR shape space $\Omega_S$, (ii) pre-trained CLIP text and image encoders $E_T$ and $E_I$, which map text $T$ and image $I$ to CLIP spaces $\Omega_T$ and $\Omega_I$, respectively, (iii) mapper $M$ with 12 fully-connected layers, each followed by a Leaky-ReLU, and (iv) decoder $D$ to generate the final shape $S$. Specifically, we use DVR (Niemeyer et al., 2020) as the SVR model when presenting our method, unless otherwise specified.

Overall, we introduce a novel two-stage feature-space-alignment approach to connect the text, image, and shape modalities. In detail, we first train CLIP2Shape mapper $M$ to connect the CLIP image space $\Omega_I$ and the shape space $\Omega_S$ from the pre-trained SVR model (see Figure 2 (a)). Then, we fine-tune $M$ at test time using a CLIP consistency loss $L_c$ to further connect the CLIP text space $\Omega_T$ with $\Omega_S$ (see Figure 2 (b)). Last, we may further optimize the texture and structure style of $S$ by fine-tuning the decoders (see Figure 2 (c)).

In the following, we first introduce two empirical studies on the CLIP feature space in Section 3.2, then present our two-stage feature-space-alignment approach in Section 3.3. Further, Section 3.4 presents our text-guided shape stylization method and Section 3.5 discusses the compatibility of our approach with different SVR models and our extension to generate a broad range of categories.

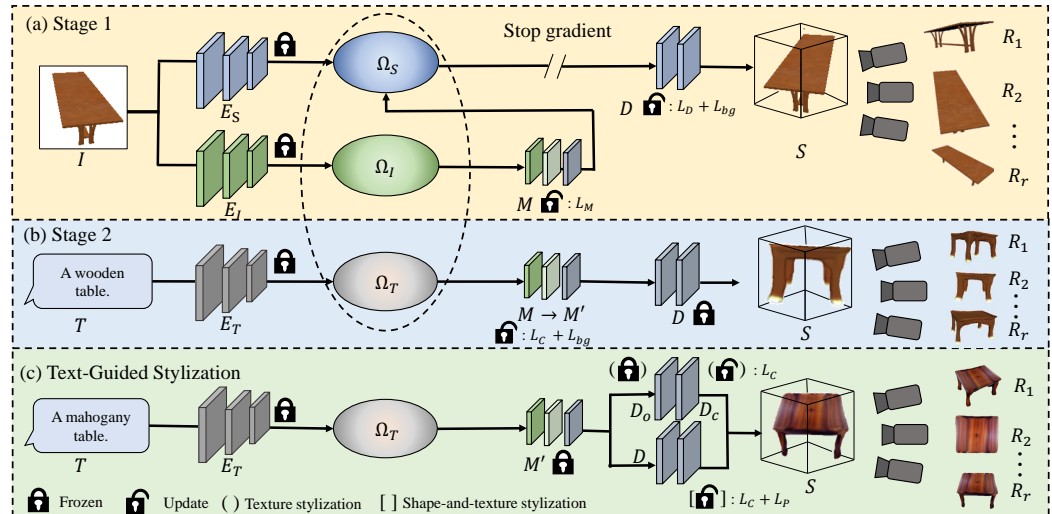

Figure 2: Overview of our text-guided 3D shape generation framework, which has three major stages. (a) Leveraging a pre-trained SVR model, in stage-1 feature-space alignment, we train the CLIP2Shape mapper $M$ to map the CLIP image feature space $\Omega_I$ to shape space $\Omega_S$ of the SVR model with $E_S$, $E_I$ frozen, and fine-tune decoder $D$ with an additional background loss $L_{bg}$. $M$ and $D$ are trained with their own losses separately at the same time by stopping the gradients from SVR loss $L_D$ and background loss $L_{bg}$ to propagate to $M$. (b) In stage-2 feature-space alignment, we fix $D$ and fine-tune $M$ into $M'$ by encouraging CLIP consistency between input text $T$ and the rendered images at test time. (c) Last, we optimize the style of the generated shape and texture of $S$ for $T$. At the inference, we use stage 2 to generate 3D shape from $T$ and (c) is optional.

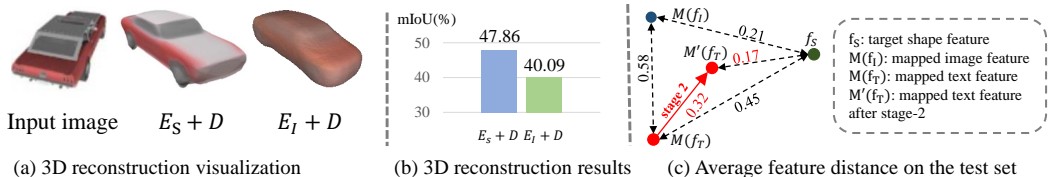

(a) 3D reconstruction visualization     (b) 3D reconstruction results     (c) Average feature distance on the test set

Figure 3: Empirical studies on the CLIP feature space for text-guided 3D shape generation.

## 3.2 EMPIRICAL STUDIES AND MOTIVATIONS

Existing works (Sanghi et al., 2022; Zhou et al., 2022; Wang et al., 2022b) mostly utilize CLIP directly without analyzing how it works and discussing its limitations. To start, we investigate the merits and drawbacks of leveraging CLIP for text-guided 3D shape generation by conducting the following two empirical studies to gain more insight into the CLIP feature space.

### 3.2.1 WHETHER CLIP FEATURE SPACE SUFFICIENTLY GOOD FOR 3D SHAPE GENERATION?

First, we study the representative capability of the CLIP image feature space $\Omega_I$ by trying to generate 3D shapes from this space. Specifically, we replace the SVR image encoder $E_S$ with the CLIP image encoder $E_I$, and optimize implicit decoder $D$ using multi-view losses like DVR (Niemeyer et al., 2020) with $E_I$ frozen. This approach can be extended to text-to-shape generation by replacing $E_I$ with CLIP text encoder $E_T$ during the inference. To compare the performance of $E_S$ and $E_I$, we evaluate 3D mIoU between their generated shapes and GTs. The results are as follows: the standard SVR pipeline $E_S+D$ achieves 47.86% mIoU while replacing the SVR encoder $E_S$ with CLIP encoder $E_I$ ($E_I+D$) degrades the performance to 40.09%. From the results and qualitative comparison shown in Figures 3 (a, b), we can see that the CLIP image space $\Omega_I$ has *inferior representative capability to capture details of the input image* for 3D shape generation. This is not surprising, since the pre-trained $E_I$ from CLIP is targeted to extract semantic-aligned features from texts rather than extracting details from images. Hence, image details relevant to 3D reconstruction are lost, *e.g.*, textures. On the contrary, $E_S$ from the SVR model is optimized for 3D generation from images, so it maintains more

necessary details. The above result motivates us to design a mapper $M$ from $\Omega_I$ to $\Omega_S$ and then generate shapes from $\Omega_S$ instead of $\Omega_I$ for better generative fidelity.

### 3.2.2 HOW THE CLIP IMAGE AND TEXT FEATURE GAP INFLUENCES 3D SHAPE GENERATION?

Second, we investigate the gap between the normalized CLIP image feature $f_I \in \Omega_I$ and normalized CLIP text feature $f_T \in \Omega_T$; (see also the CLIP image and text feature spaces in Figure 1 (a)) and how such gap influences text-guided 3D shape generation. Specifically, we randomly sample 300 text-shape pairs from the text-shape dataset (Chen et al., 2018), then evaluate the cosine distance between $f_I$ and $f_T$, *i.e.*, $d = 1 - \text{cosine\_similarity}(f_I, f_T)$, where $f_I$ is the CLIP feature of the rendered images from the corresponding shape. We repeat the experiment and obtain $d(f_T, f_I) = 0.783 \pm 0.004$. The result reveals *a certain gap between the CLIP text and image features in this dataset, even though they are paired.* Also, the angle in the feature space between the two features is around $\arccos(1 - 0.783) = 1.35$ rad in this dataset (Chen et al., 2018). Having said that, directly replacing $f_I$ with $f_T$ like Sanghi et al. (2022); Zhou et al. (2022) in inference may harm the consistency between the output shape and the input text. As demonstrated in Figure 3 (c), directly replacing $f_I$ with $f_T$ causes a cosine distance of 0.45 to $f_S \in \Omega_S$ (see Figure 3 (c)), which is significantly larger than the distance between $M(f_I)$ and $f_S$ (0.21). Our finding is consistent with the findings in Liang et al. (2022). It motivates us to further fine-tune $M$ into $M'$ at test time, such that we can produce feature $M'(f_T)$, which is closer to $f_S$ than $M(f_T)$.

### 3.3 TWO-STAGE FEATURE-SPACE ALIGNMENT

Following the above findings, we propose a two-stage feature-space-alignment approach to first connect image space $\Omega_I$ and shape space $\Omega_S$ and further connect text space $\Omega_T$ to shape space $\Omega_S$ with the image space $\Omega_I$ as the stepping stone.

**Stage-1 alignment: CLIP image-to-shape mapping.** Given multi-view RGB or RGBD images for training, the stage-1 alignment is illustrated in Figure 2 (a). Considering that shape space $\Omega_S$ contains richer object details than the image space $\Omega_I$, while $\Omega_I$ provides a joint text-image embedding with the input text space $\Omega_T$, we introduce a fully-connected CLIP2Shape mapper $M$ to map image feature $f_I$ to shape space $\Omega_S$. Taking a rendered image $I$ as input, $M$ is optimized with an $L_2$ regression between $M(f_I)$ and standard SVR feature $f_S = E_S(I)$ according to Equation (1) below:

$$L_M = \sum_{i=1}^{N} ||M(f_{I,i}) - E_S(I_i)||_2^2 \tag{1}$$

where $N$ is the number of images in the training set and $f_{I,i}$ is the normalized CLIP feature of $I_i$.

Also, we fine-tune decoder $D$ to encourage it to predict a white background, which helps the model to ignore the background and extract object-centric feature (see Figure 4), while maintaining its 3D shape generation capability. To this end, we propose a new background loss $L_{\text{bg}}$ in Equation (2) below to enhance the model's foreground object awareness to prepare for the second-stage alignment.

$$L_{\text{bg}} = \sum_{p} ||D_c(p) - 1||_2^2 \mathbb{1}(\text{ray}(o, p) \cap F = \emptyset) \tag{2}$$

where $\mathbb{1}$ is the indicator function; $F = \{p : D_o(p) > t\}$ indicates the foreground region, in which the occupancy prediction $D_o(p)$ is larger than threshold $t$; $p$ is a query point; $\text{ray}(o, p) \cap F = \emptyset$ means the ray from camera center $o$ through $p$ does not intersect the foreground object marked by $F$; and $D_c(p)$ is the color prediction at query point $p$. In a word, $L_{\text{bg}}$ encourages $D$ to predict the background region as white color (value 1), such that $E_I$ can focus on and better capture the foreground object. In addition, to preserve the 3D shape generation capability of $D$, we follow the loss $L_D$, with which $D$ has been optimized in the SVR training. In this work, we adopt Niemeyer et al. (2020).

Hence, the overall loss in stage-1 training is $\lambda_M L_M$ for mapper $M$ and $\lambda_{\text{bg}} L_{\text{bg}} + L_D$ for decoder $D$, where $\lambda_M$ and $\lambda_{\text{bg}}$ are weights. The stage-1 alignment provides a good initialization for the test-time optimization of stage 2.

**Stage-2 alignment: text-to-shape optimization.** Given a piece of text, the stage-2 alignment aims to further connect the text and shape modalities. Specifically, it searches for shape $S$ that best matches

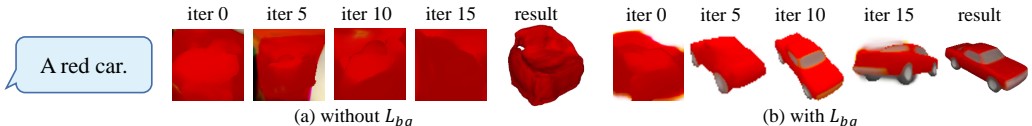

Figure 4: Effect of generating shapes from the same text with/without background loss $L_{\text{bg}}$.

the input text $T$. To this end, we formulate a fast test-time optimization to reduce the gap between the text and image CLIP features $f_T$ and $f_I$, as discussed earlier in the second empirical study.

As shown in Figure 2 (b), given input text $T$, we replace image encoder $E_I$ with text encoder $E_T$ to extract CLIP text feature $f_T$, then fine-tune $M$ with CLIP consistency loss between input text $T$ and $m$ images $\{R_i\}_{i=1}^m$ rendered with random camera poses from output shape $S$; see Equation 3:

$$L_C = \sum_{i=1}^m \langle f_T \cdot \frac{E_I(R_i)}{||E_I(R_i)||}\rangle \qquad (3)$$

where $\langle \cdot \rangle$ indicates the inner-product.

In stage-2 alignment, we still adopt $L_{\text{bg}}$ to enhance the model's foreground awareness. Comparing Figures 4 (a) and (b), we can see that the stage-2 alignment is able to find a rough shape with $L_{\text{bg}}$ in around five iterations, yet failing to produce a reasonable output without $L_{\text{bg}}$, since having the same color prediction on both foreground and background hinders the object awareness of the model.

Thanks to the joint text-image embedding of CLIP, the gap between text feature $f_T$ and shape feature $f_S$ has already been largely narrowed by $M$. Therefore, the stage-2 alignment only needs to fine-tune $M$ with 20 iterations using the input text, taking only around 85 seconds on a single GeForce RTX 3090 Ti, compared with 72 minutes taken by Dream Fields (Jain et al., 2022) at test time. After this fine-tuning, we can readily obtain a plausible result; see, *e.g.*, the "result" shown in Figure 4 (b). Our ISS is a novel and efficient approach for 3D shape generation from text.

**Diversified generation.** In general, shape generation from text is one-to-many. Hence, we further extend our approach with diversified 3D shape generation from the same piece of input text. Unlike the existing works, which require additional and complex modules, *e.g.*, GANs (Chen et al., 2018), IMLE (Liu et al., 2022), and normalizing flow network (Sanghi et al., 2022), we can simply perturb the image and text features for diversified generation. Specifically, after stage-1 alignment, we randomly permute $f_I$ as an initialization and $f_T$ as the ground truth by adding normalized Gaussian noises $z_1 = h_1/||h_1||, z_2 = h_2/||h_2||$, where $h_1, h_2 \sim N(0,1)$ to derive diversified features

$$\hat{f}_I = \tau_1 f_I + (1 - \tau_1)z_1 \quad \text{and} \quad \hat{f}_T = \tau_2 f_T + (1 - \tau_2)z_2, \qquad (4)$$

where $\tau_1, \tau_2$ are hyperparameters to control the degrees of permutation. With permuted $\hat{f}_I$ and $\hat{f}_T$ in stage-2 alignment, our model can converge to different 3D shapes for different noise.

## 3.4 TEXT-GUIDED STYLIZATION

The two-stage feature-space alignment is already able to generate plausible 3D shapes; see, *e.g.*, Figures 2 (b) and 4 (b). However, the generative space is limited by the representation capability of the employed SVR model, *e.g.*, DVR (Niemeyer et al., 2020) can only generate shapes with limited synthetic patterns as those in ShapeNet. However, a richer and wider range of structures and textures are highly desired. To this end, we equip our model with a text-guided stylization module to enhance the generated shapes with novel structure and texture appearances, as shown in Figures 2 (c) and 1.

Specifically, for texture stylization, we first duplicate $D$ (except for the output layer) to be $D_o$ and $D_c$, then put the output occupancy prediction layer and color prediction layer on top of $D_o$ and $D_c$, respectively. Further, we fine-tune $D_c$ with the same CLIP consistency loss as in Equation (3), encouraging the consistency between input text $T$ and the $m$ rendered images $\{R_i\}_{i=1}^m$.

Besides textures, novel structures are also desirable for shape stylization. Hence, we further incorporate a shape-and-texture stylization strategy to create novel structures. To enable shape sculpting, we fine-tune $D$ with the same CLIP consistency loss in Equation 3. At the same time, to maintain the

overall structure of the initial shape $S$, we propose a 3D prior loss $L_P$ shown in Equation (5), aiming at preserving the 3D shape prior learned by the two-stage feature-space alignment.

$$L_{\mathrm{P}} = \sum_p |D_o(p) - D'_o(p)| \tag{5}$$

where $p$ is the query point, and $D_o$, $D'_o$ are the occupancy predictions of the initial $D$ and the fine-tuned $D$ in the stylization process, respectively. To improve the consistency between the generated texture and the generated shape, we augment the background color of $R_i$ with a random RGB value in each iteration. Please find more details in the supplementary material.

### 3.5 COMPATIBILITY WITH DIFFERENT SVR MODELS

Besides DVR (Niemeyer et al., 2020), our ISS framework is compatible with different SVR models. For example, we can adapt it with the most recent SVR approach SS3D (Alwala et al., 2022) that leverages in-the-wild single images for 3D generation. With this model, our framework can generate a wider range of shape categories by using SS3D's encoder and decoder as shape encoder $E_S$ and decoder $D$ in our framework, respectively. Here, we simply follow the same pipeline as in Figure 2 to derive a text-guided shape generation model for the in-the-wild categories; see our results in Section 4.4. Notably, we follow the losses in Alwala et al. (2022) in place of $L_D$ (see Section 3.3) in stage-1 training, requiring only single-view images without camera poses. More importantly, our approach's high compatibility suggests that it is orthogonal to SVR, so its performance can potentially be further upgraded with more advanced SVR approaches in the future.

## 4 EXPERIMENTS

### 4.1 DATASETS, IMPLEMENTATION DETAILS, AND METRICS

With multi-view RGB or RGBD images and camera poses, we can train ISS on the synthetic dataset ShapeNet (Chang et al., 2015) (13 categories) and the real-world dataset CO3D (Reizenstein et al., 2021) (50 categories). To evaluate our generative performance, we create a text description set with four texts per category on ShapeNet and two texts per category on CO3D. SS3D (Alwala et al., 2022) takes single-view in-the-wild images in training; as their data has not been released, we only evaluate our method on some of their categories. To evaluate the performance, we employ Fréchet Inception Distance (FID) (Heusel et al., 2017), Fréchet Point Distance (FPD) (Liu et al., 2022) to measure shape generation quality, and conduct a human perceptual evaluation to further assess text-shape consistency. Please refer to the supplementary material for more details on the metrics and implementation details.

### 4.2 COMPARISONS WITH STATE-OF-THE-ART METHODS

We compare our approach with existing works (Sanghi et al., 2022; Jain et al., 2022) both qualitatively and quantitatively. For a fair comparison, we use their official codes on GitHub to generate shapes on our text set. Table 1 shows quantitative comparisons, whereas Figure 5 shows the qualitative comparisons. Comparing existing works and ours in Table 1, we can see that our approach outperforms two state-of-the-art works by a large margin for both generative quality and text-shape consistency scores. On the other hand, the qualitative comparisons in Figure 5 (a,b) show that CLIP-Forge (Sanghi et al., 2022) produces low-resolution shapes without texture, and some generated shapes are inconsistent with the input text, *e.g.*, "a wooden boat." Dream Fields (Jain et al., 2022) cannot generate reasonable shapes from the input text on the top row and its generated shape on the bottom row is also inconsistent with the associated input text. On the contrary, our approach (Figure 5 (i)) can generate high-fidelity shapes that better match the input text. Note that we only utilize two-stage feature-space alignment without stylization in producing our results. Please refer to the supplementary file for more results and visual comparisons.

### 4.3 ABLATION STUDIES

To manifest the effectiveness of our approach, we conduct ablation studies on the following baselines (see Table 1 and Figure 5): generate shapes from $\Omega_I$ ($E_I + D$), optimize stage-2 alignment without stage-1 (w/o stage 1), conduct stage-1 alignment without stage-2 (w/o stage 2), disable the background loss in stage 1, stage 2 and both (w/o $L_{\mathrm{bg\_1}}$, w/o $L_{\mathrm{bg\_2}}$, w/o $L_{\mathrm{bg}}$), and two additional baselines that first create images and then 3D shapes (GLIDE+DVR, LAFITE+DVR). More details on the setup and analysis are provided in the supplementary material Section 3. Our results outperform all the existing works and baselines in terms of fidelity and text-shape consistency by a large margin.

Table 1: Quantitative comparisons with existing works and baselines.

| Method Type | Method | FID ($\downarrow$) | Consistency Score (%) ($\uparrow$) | FPD ($\downarrow$) | A/B/C Test |
|---|---|---|---|---|---|
| Existing works | CLIP-Forge | 162.87 | $41.83 \pm 17.62$ | 37.43 | $8.90 \pm 4.12$ |
| | Dream Fields | 181.25 | $25.38 \pm 12.33$ | N.A. | N.A. |
| Ablation Studies | $E_I+D$ | 181.88 | $20.97 \pm 13.59$ | 38.61 | N.A. |
| | w/o Stage 1 | 222.96 | $1.92 \pm 2.22$ | 79.41 | N.A. |
| | w/o Stage 2 | 202.33 | $29.52 \pm 14.86$ | 41.71 | N.A. |
| | w/o $L_{\text{bg\_1}}$ | 149.45 | $29.45 \pm 14.67$ | 40.85 | N.A. |
| | w/o $L_{\text{bg\_2}}$ | 156.52 | $31.55 \pm 8.87$ | 38.31 | N.A. |
| | w/o $L_{\text{bg}}$ | 178.34 | $30.96 \pm 15.49$ | 40.98 | N.A. |
| Text2Image+SVR | GLIDE+DVR | 212.41 | $8.85 \pm 7.94$ | 41.33 | N.A. |
| | LAFITE+DVR | 135.01 | $52.12 \pm 11.05$ | 37.55 | $11.70 \pm 4.11$ |
| Ours | ISS | $\mathbf{124.42 \pm 5.11}$ | $\mathbf{60.0 \pm 10.94}$ | $\mathbf{35.67 \pm 1.09}$ | $\mathbf{21.70 \pm 5.19}$ |

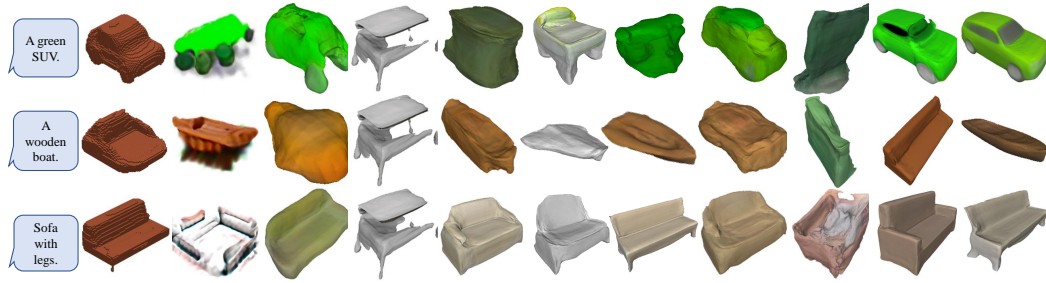

(a) CLIP-Forge  (b) Dream Fields  (c) $E_{CLIP\_I} + D$  (d) w/o stage-1  (e) w/o stage-2  (f) w/o $L_{bg\_1}$  (g) w/o $L_{bg\_2}$  (h) w/o $L_{bg}$  (i) GLIDE+DVR  (j) Lafite + DVR  (k) Ours

Figure 5: Qualitative comparisons with existing works and baselines.

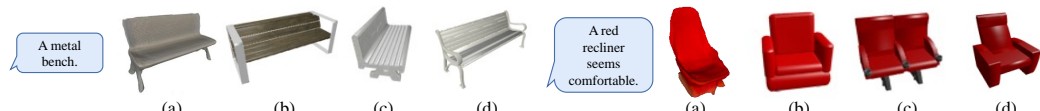

Figure 6: Our approach is able to generate novel shapes, not in the training set. (a) shows our results and (b,c,d) are the top-three shapes retrieved from the training set.

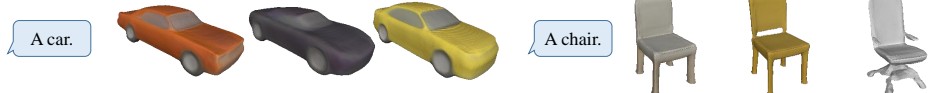

Figure 7: Our approach can generate diversified results from the same input text.

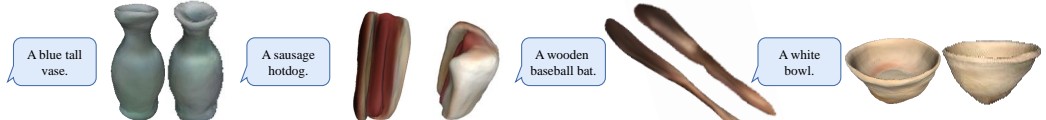

Figure 8: Results on CO3D dataset. We show two different views of each result.

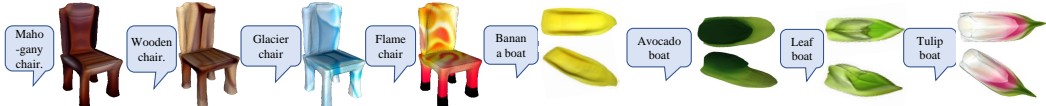

Figure 9: Text-guided stylization. Left: texture stylization. Right: shape-and-texture stylization.

## 4.4 MORE ANALYSIS ON GENERATIVE RESULTS OF ISS

Next, we present evaluations on the generative novelty and diversity, as well as the scalability of our two-stage feature-space alignment. Then, we show more text-guided stylization results and how our ISS approach generalizes to a wide range of categories and generates shapes with better fidelity.

**Generation novelty.** Our approach is able to generate novel shapes beyond simple retrieval from the training data. As shown in Figure 6, from the input text, we first generate our result in (a) and then take our generated shape to retrieve the top-three closest shapes in the associated training set

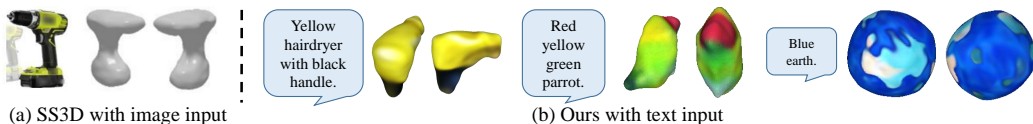

(a) SS3D with image input                    (b) Ours with text input

Figure 10: With single images for training (without camera poses), our approach can produce results for a broad range of categories, by adopting (Alwala et al., 2022). Two different views are rendered.

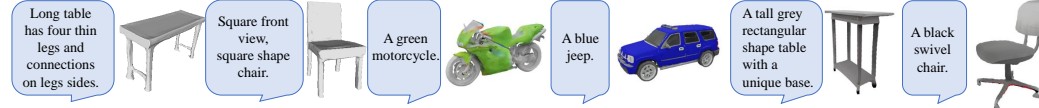

Figure 11: Results of ISS built upon the SVR model of IM-Net (left two) and GET3D (right four).

based on the cosine similarity between the CLIP features of the rendered images $f_I$ and input text $f_T$ as the retrieval metric. Our results with the two-stage feature-space alignment are not in the training sets, showing that our ISS approach can generate novel shapes beyond the training set, even without stylization. It is not surprising, as our approach shares the generative space with the pre-trained SVR model and can potentially generate all shapes that the pre-trained SVR model can produce.

**Generation diversity.** By perturbing the features and injecting randomness on initialization, ISS is able to generate diversified results from the same text input. As shown in Figure 7, ISS produces various cars and chairs from the same piece of text. Quantitative results are presented in the supplementary material Section 3.4 where our method achieves a high PS (Point Score).

**Generation fidelity.** To assess the capability of ISS in generating realistic real-world 3D shapes, we train the SVR model on the CO3D dataset (Reizenstein et al., 2021) which is a real-world dataset, and ISS leverages the learned feature space for text to shape generation without paired data. As shown in Figure 8, our model is able to generate real-world shapes. As far as know, this is first work that investigates text-guided shape generation and on real-world datasets can generate realistic 3D shapes.

**Generation beyond the capability of SVR model.** Our text-guided stylization module equips our model with the capability to generate 3D shapes beyond the SVR model. As shown in Figure 9 and Figure 2 (c), our model is able to create realistic and fantasy novel structures and textures that match text descriptions. Please refer to the supplementary material Section 4 for more details.

**Generality and Scalability of ISS on other SVR models.** Our model is generic and can work together with other SVR models. To evaluate the generality and scalablity of our model, we employ SS3D (Alwala et al., 2022), IM-Net (Chen & Zhang, 2019), and GET3D (Gao et al., 2022) as SVR models to provide the feature space. It is worth noting that SS3D is capable of generating shapes of more categories and IM-Net, GET3D can produce high fidelity results. First, as shown in Figure 10, built upon SS3D, our approach can generate shapes of more real-world categories, *e.g.*, bird. Note that our model can generate shapes with comparable or even better qualities compared with initial SS3D model that takes an image as input. Second, when combined with IM-Net and GET3D, our model can fully exploit their generative capabilities and produces high-quality 3D shapes as shown in Figure 11. The above manifests that ISS is generic and compatible to advanced models for generating shapes of more categories and higher qualities.

## 5 CONCLUSION

In this paper, we present a novel approach for text-guided 3D shape generation by leveraging the image modality as a stepping stone. Leveraging the joint text-image embeddings from CLIP and 3D shape priors from a pre-trained SVR model, our approach eliminates the need for the paired text and shape data. Technically, we have the following contributions. First, we step-by-step reduce the semantic gap among the text, image and shape modalities through our two-stage feature-space alignment approach. Second, our text-guided stylization technique effectively enriches our generated shapes with novel structures and textures in various styles. Third, our approach is compatible with various single-view reconstruction approaches and can be further extended to generate a wide range of categories with only single images without camera poses in training. Experiments on ShapeNet, CO3D, and multiple single-image categories manifest the superiority of our framework over the two state-of-the-art methods and various baselines. Limitations are discussed in the supplementary files.

## ACKNOWLEDGEMENTS

The work has been supported in part by the Research Grants Council of the Hong Kong Special Administrative Region (Project no. CUHK 14206320), General Research Fund of Hong Kong (No. 17202422), Hong Kong Research Grant Council - Early Career Scheme (Grant No. 27209621), and National Natural Science Foundation of China (No. 62202151).

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
