# OpenReview forum: "ISS: Image as Stepping Stone for Text-Guided 3D Shape Generation"
_ICLR.cc/2023/Conference — ICLR 2023 notable top 25%_

### Official Review · Reviewer_YJ7a · 2022-10-23

**Confidence:** 3
**Correctness:** 2
**Technical Novelty And Significance:** 2
**Empirical Novelty And Significance:** 3
**Recommendation:** 6

**Clarity, Quality, Novelty And Reproducibility:**

The quality of the writing is overall good. The quality/originality of the proposed method is also good. Most of the paper is clearly clarified. Some detailed settings are unclear, as suggested in the weaknesses.

**Strength And Weaknesses:**

Pros

-This paper is well organized and easy to follow.

-The proposed method is simple and effective. The alignment of the text feature space to the shape feature space through the intermediate image feature space avoids the requirement of paired text-shape training data. The introduced losses, especially background loss in Eq. (3) and training tricks, like diversified generation and augmentation make this model work.

-The experiments cover the comparison with existing methods and ablation studies. The experimental results show better performance than existing methods with numerical results and visual examples.

Cons
-Figure 2 is unclear. The authors divided the figure into three subfigures, which corresponds to different stages of training. However, from the figure, it is unclear which part of the model is trained, what loss is applied and how the training is performed.

-In Section 3.2.1, the authors use the mIoU metric to measure the 3D reconstruction results and observe a degradation by replacing the SVR encoder with the CLIP encoder. However, it is unclear how the added mapper can help to solve the problem.

-In Section 3.3, the authors claimed ' we fine-tune decoder D ...'. It is unclear the mapper M and decoder D are trained together or separated at the first stage.

-It is unclear how the camera poses are set and selected during training, both the first stage and second stage.

-It is unclear how the decoder D is duplicated to D_o and D_c in Section 3.4 and Figure 2.

-The selected metric FID may be unfair to the compared method, especially CLIP-Forge as it only supports the color generation for shapes. This can be found in Figure 5. A feasible solution is to add the metric of Maximum measure distance (MMD) as in CLIP-Forge along with FID to compare different methods. Furthermore, the pretrained model used in FID is not clarified.

-There is no Figure 8(b), which is stated in 'Generation beyond the capability of SVR model'.

**Summary Of The Paper:**

In this paper, the authors propose a text-to-3D shape generation method which does not required paired text-shape training data. This work exploits the pretrained CLIP feature spaces for text and images and a pre-trained single-view reconstruction (SVR) model. The text feature space is aligned to an intermediate image feature space and then the target shape feature space. Various losses are designed to ensure the alignments. A further shape stylization module is introduced to generate new structures and textures for the generated shape. Experiments are conducted with different SVR models on ShapeNet and CO3D datasets. Experimental results show the proposed method can achieve higher performance than existing methods in terms of FID and FPD.

**Summary Of The Review:**

The idea and implementation put this paper above the borderline. The unclear setting of the model and unsatisfied experimental comparison degrade the quality of this paper. Overall, I vote for a positive score.

---

> ### Author Response · Authors · 2022-11-18
> **Author feedback to Reviewer YJ7a Part-2**
>
>
>
> **Q6**: The selected metric FID may be unfair to the compared method, especially CLIP-Forge as it only supports the color generation for shapes. This can be found in Figure 5. A feasible solution is to add the metric of Maximum measure distance (MMD) as in CLIP-Forge along with FID to compare different methods. Furthermore, the pre-trained model used in FID is not clarified.
>
> **A6**: Thanks for your question. Our FPD metric in Table 1 of the main paper is already designed to measure the shape quality without color; see Figure 3 in the supplementary material for some input examples. To measure MMD, we still adopt the above point clouds in Figure 3 without color. Following [2], we extract the point cloud features with a pre-trained PointNet model and measure the MMD between the generated shapes with the ground-truth ones. The smaller the MMD value the closer the sample to the ground truth. We then measure MMD for the results of both CLIP-Forge and ours: **0.291** (CLIP-Forge) and **0.260±0.01** (ours), which further demonstrate that our generated results outperform the results of CLIP-Forge even without considering the color.
>
> As for FID, we adopt the official model with Inception Net trained on ImageNet, which is widely used to evaluate generative quality and realism. We do not train a model on ShapeNet, since it is too small to train a better network for evaluating the FID than models trained on ImageNet.
>
> In addition, we randomly sample **2600** images in the ShapeNet dataset as ground truths for FID evaluation instead of using images from ImageNet. It helps to measure the similarity between the generated shapes and ground truths of ShapeNet.
> We have added the details in Section 2 “Metrics” in the supplementary material.
>
> [2] Liu, Zhengzhe, et al. "Towards Implicit Text-Guided 3D Shape Generation." Proceedings of the IEEE/CVF Conference on Computer Vision and Pattern Recognition. 2022.
>
> **Q7**: There is no Figure 8(b), which is stated in 'Generation beyond the capability of SVR model'.
>
> **A7**: Thanks for your careful review.  Sorry for the typo. It should be Figure 9.  We have revised it in the updated manuscript.

---

> ### Author Response · Authors · 2022-11-18
> **Author Feedback to Reviewer YJ7a Part-1**
>
> **Q1**: Figure 2 is unclear. The authors divided the figure into three subfigures, which corresponds to different stages of training. However, from the figure, it is unclear which part of the model is trained, what loss is applied and how the training is performed.
>
> **A1**: Thanks for your question. We have updated Figure 2 to show more details on which losses are used in the training of different parts. Specifically, in stage 1, decoder D is updated with the original SVR loss L_D and the background loss L_{bg}, while mapper M is updated with the regression loss L_M. In stage 2, mapper M is updated with the CLIP consistency loss. In stage 3, decoder D (or D_c) is updated with the CLIP consistency loss.
>
> **Q2**: In Section 3.2.1, the authors use the mIoU metric to measure the 3D reconstruction results and observe a degradation by replacing the SVR encoder with the CLIP encoder. However, it is unclear how the added mapper can help to solve the problem.
>
> **A2**: Thanks for your question.
>
> As discussed in Section 3.2.1 of the main paper, the CLIP image encoder cannot fully capture the image details relevant to 3D reconstruction, but the SVR encoder maintains more necessary details. This motivates us to generate shapes from \Omega_S instead of \Omega_I for better generative fidelity. With the mapper, we can map a feature from \Omega_I to \Omega_S, and with the stage-2 feature space alignment, we further finetune the mapper M to search for a good feature f_S in \Omega_S, instead of \Omega_I, that matches the text and generate shapes with good quality.
>
> **Q3**: In Section 3.3, the authors claimed ' we fine-tune decoder D ...'. It is unclear the mapper M and decoder D are trained together or separated at the first stage.
>
> **A3**: We would like to clarify that mapper M and decoder D are trained at the same time but **separately**. As shown in Figure 2 of our revised paper, we finetune the SVR decoder D with the original SVR loss L_D (to preserve the 3D shape generation capability) and an additional background loss L_{bg} (to encourage the rendered image to have white background). At the same time, we train mapper M with the regression loss L_M. To summarize, the decoder D and mapper M are trained at the same time but **separately**, and we achieve this by **stopping the gradient** between M and D to avoid the effects of L_D and L_{bg} on M.
>
> We have added the above analysis in Section 9 of the supplementary material.
>
> **Q4**: It is unclear how the camera poses are set and selected during training, both the first stage and second stage.
>
> **A4**: Thanks for your question.
>
> In Stage 1, we follow DVR [1] to set the camera poses to encourage the background to be white. Specifically, we randomly sample the distance of the camera and the viewpoint on the northern hemisphere.
>
> In Stage 2, compared with DVR [1], we sample the camera distance to be 1.5 times further compared with DVR [1]. It helps to encourage sampling more global views instead of only local ones, so that the CLIP image encoder can capture the whole shape and yield a better CLIP feature.
>
> In Stage 3, we also sample the camera distance to be 1.5 times. Since this stage aims to generate textures instead of searching for a target shape like Stages 1 and 2, only sampling view points on the northern hemisphere of the view space cannot ensure good generation quality in the bottom regions. Thus, we further randomly sample viewpoints on the southern hemisphere for random 10% training iterations to encourage the stylized results to be consistent with the text in various viewpoints.
>
> We have added the above analysis in Section 2 “Details on camera poses” of the supplementary material.
>
> [1] Niemeyer, Michael, et al. "Differentiable volumetric rendering: Learning implicit 3d representations without 3D supervision." Proceedings of the IEEE/CVF Conference on Computer Vision and Pattern Recognition. 2020.
>
>
> **Q5**: It is unclear how the decoder D is duplicated to D_o and D_c in Section 3.4 and Figure 2.
>
> **A5**: Except for the output layer, we simply duplicate them to be D_o and D_c. For the output layer, it takes d-dimensional features as input and outputs one value for occupancy and three values for RGB. We then copy dx1 weights to D_o and copy dx3 weights to D_c. See Figure 4 in the supplementary material for more details.
> We further included the above analysis in Section 2 “Decoder duplication” of the supplementary material.

---

### Official Review · Reviewer_CfJ1 · 2022-10-24

**Confidence:** 3
**Correctness:** 4
**Technical Novelty And Significance:** 2
**Empirical Novelty And Significance:** 3
**Recommendation:** 6

**Clarity, Quality, Novelty And Reproducibility:**

The article describes the structure of its method clearly and provides many experiments results, including ablation and SOTA results, to prove the effectiveness of the method. Although the method depends on the CLIP model to use image space as a step stone to transfer the text feature into shape space, the method provides a novel work to complete Text-shape task.

**Strength And Weaknesses:**

Strengths:
1.	The method does not require a large amount of data in pairs as a reference
2.	Due to powerful CLIP as the base, the method is able to obtain a convincing result with less time cost comparing to other existing work.
3.	Further, the method provides a style transfer module to enrich the result.

Weaknesses:
1.	The final step is based on SVR method, so how to ensure that the shapes obtained have good details? As shown in the experiment part, the image does not have a lot of fine details.
2.	Although using more and richer texts to obtain the final model, it seems that the method cannot generate results that not in the training domain.


**Summary Of The Paper:**

The paper proposes a new method for text-shape transfer, namely ISS(Image as Stepping stone), based on powerful CLIP, without the need of paired text-shape dataset. The ISS consists of 2 stages:
1. stage 1, the method trains a CLIP2Shape mapper to map the CLIP image features to a pre-trained detail-rich 3D shape space.
2. stage2, due to further finetune the mapper for better consistency between the text and generated shape.
3. To further enrich the generated shape according the input text, the method proposes a style transfer module to transfer the texture to the shape.


**Summary Of The Review:**

This article proposes a method ISS based on CLIP that can effectively implement text-shape conversion, and provides a large number of credible experimental results as an illustration.

---

> ### Author Response · Authors · 2022-11-18
> **Author Feedback to Reviewer CfJ1**
>
> **Q1**: The final step is based on the SVR method, so how to ensure that the shapes obtained have good details? As shown in the experiment part, the image does not have a lot of fine details.
>
> **A1**: Thanks for your comment.
>
> The generative quality of our model is limited by the SVR model we built upon, as discussed in the Limitation section in the supplementary material.
>
> Our method is compatible with 3D generative models to leverage the advancements in this area. To this end, we further adopt the very recent state-of-the-art 3D generative approach GET3D [1] to work with our approach. When working with GET3D, our approach can generate 3D shapes with fine details from texts, as shown in Figure 11 of the main paper and Figure 11 of the supplementary material.
>
> Besides, this encouraging results with GET3D further demonstrate the high compatibility of our approach to work with different SVR models. Our method can potentially achieve better performance when working with other SVR models in the future.
>
> [1] Gao, Jun, et al. "GET3D: A Generative Model of High Quality 3D Textured Shapes Learned from Images." NeurIPS 2022.
>
>
> **Q2**: Although using more and richer texts to obtain the final model, it seems that the method cannot generate results that not in the training domain.
>
> **A2**: With our text-guided 3D shape stylization, our approach can generate results that are not in the training domain. See the results shown in Figure 9 (main paper) and Figure 7 (supplementary material).
>
> We train our model on the ShapeNet dataset (https://shapenet.org/). This dataset does not contain shapes similar to our “glacier chair”, “banana boat”, “snow mountain sofa”, “rose chair", etc. Therefore, our stylized results are not in the training domain.

---

### Official Review · Reviewer_Z3WY · 2022-10-25

**Confidence:** 5
**Correctness:** 3
**Technical Novelty And Significance:** 2
**Empirical Novelty And Significance:** 2
**Recommendation:** 6

**Clarity, Quality, Novelty And Reproducibility:**

The paper reads well. The quality of the presentation is good. The problem is interesting. However, given that the framework is very complicated, it is unclear whether the paper is reproducible without making the code publicly available to the community.

**Strength And Weaknesses:**

Strengths:
1. The problem is very interesting.
2. The idea of using images to connect two modalities is good.
3. Results are good compared to existing methods

Weaknesses:
1. The proposed framework shown in Figure 2 is too complicated. This means that there are many learnable parameters in the framework. When reporting results with comparisons to other methods, model complexity should also be taken into consideration. For example, when looking at Table 1, it is unclear whether the performance gain over the competing methods is from designing the "right" algorithm or is it simply from using more learnable parameters.
2. In Table 1, what does Ours mean in the category column?
3. The paper claims that the approach can stylize shapes with realistic structures and textures. However, the green SUV example in Figure 5 and the results in Figure 9 don't look realistic.

**Summary Of The Paper:**

This paper tackles the problem of text-guided 3D shape generation. This is a challenging task due to the absence of large paired text-shape datasets. This paper proposes a method called Image as Stepping Stone (ISS) that introduces 2D images as stepping stones to connect the two modalities, which in turn avoids the need for paired data. Experimental results show some improvement over existing methods.

**Summary Of The Review:**

Please see comments in the two boxes above.

---

> ### Author Response · Authors · 2022-11-18
> **Author Feedback to Reviewer Z3WY  Part-2**
>
> **Q2**: In Table 1, what does Ours mean in the category column?
>
> **A2**: The “Category” column means a summary or description of the set of methods of column two “Method”. “Ours” means our method “ISS”. We have revised it to be “Method Type”.
>
> **Q3**: The paper claims that the approach can stylize shapes with realistic structures and textures. However, the green SUV example in Figure 5 and the results in Figure 9 don't look realistic.
>
> **A3**: Please note that the major module responsible for enhancing realisticness is stage 3 (Text-guided stylization). However, all the shapes in Figure 5 are output from stage-2. The reason for the lack of realisticness of stage 2 outputs is explained below. First, the original SVR used to generate shape is trained on ShapeNet, which is a synthetic dataset with unrealistic images. To address this issue, when we train our model on CO3D (see Figure 8 in the main paper), we can generate better results in terms of realism. Second, the generative quality of our model is limited by the SVR model we built upon, as discussed in the Limitation section in the supplementary file.
>
> With the stylization module, the model has more capability to generate realistic 3D shapes (see Figure 9 mahogany chair and wooden chair). Besides, we can generate fantasy styles that do not exist in the real world (see Figure 9 glacier chair and flame chair). Admittedly, the pre-trained SVR model and its ability still pose a constraint on the quality of 3D shape generation and their realisticness.
>
> Further, our model can also benefit from existing 3D generation models to generate realistic 3D shapes. Here, we experiment with the most recent work GET3D [1]. By leveraging its 3D shape feature space, we train our model to enable text-guided 3D shape generation. The results are shown in Figure 11 in the paper and Figure 11 in the supplementary material. We can output realistic 3D shapes working together with GET3D.  The encouraging results with GET3D further demonstrate the high compatibility of our approach with different SVR models.  Our method can potentially achieve even better performance with other SVR models in the future.
>
> [1] Gao, Jun, et al. "GET3D: A Generative Model of High Quality 3D Textured Shapes Learned from Images." NeurIPS 2022.
>
> **Q4**: Given that the framework is very complicated, it is unclear whether the paper is reproducible without making the code publicly available to the community.
>
> **A4**: First, as stated at the end of the Introduction section, we WILL RELEASE our codes and trained models upon the publication of this work.
>
> Second, as decoder D and mapper M are trained separately (see the answer to Q1), how they contribute to our final result is very clear.
>
> (i) Our framework is lightweight and flexible.
>
> (ii) Our framework outperforms existing works that have more parameters (CLIP-Forge) and more training time (Dream Field) quantitatively in terms of generative quality and text-shape consistency; see the quantitative comparison in Table 1 of the main paper.
>
> (iii) We show the extensibility and generality of our model by combining it with several 3D generation models, including SS3D, IM-Net, and GET3D. This manifests that our proposed pipeline generally works well.

---

> > ### Comment · Reviewer_Z3WY · 2022-11-22
> > **Post rebuttal comments**
> >
> > I have read the author's response and am satisfied with the clarifications. I'd like to raise my rating to 6.

---

> > > ### Author Response · Authors · 2022-11-22
> > > **Thanks for your comments!**
> > >
> > > Thanks for your comments! Your review is very helpful for our paper.

---

> ### Author Response · Authors · 2022-11-18
> **Author Feedback to Reviewer Z3WY  Part-1**
>
> **Q1**: The proposed framework shown in Figure 2 is too complicated. This means that there are many learnable parameters in the framework. When reporting results with comparisons to other methods, model complexity should also be taken into consideration. For example, when looking at Table 1, it is unclear whether the performance gain over the competing methods is from designing the "right" algorithm or is it simply from using more learnable parameters.
>
> **A1**: Thanks for your comment.
>
> We have redrawn Figure 2 to make it clearer. Given a text description, our approach maps the text feature f_T into a shape feature space f_s of an SVR model, and then the decoder of the SVR model is used to produce the shape. To that end, our proposed framework contains three stages: stage 1: learn a mapper M to map image feature/text feature into SVR 3D feature; stage 2: optimize M to deliver M’, which further reduces the gap between text feature and 3D SVR feature; stage 3 text-guided stylization finetune the decoder D. Our model incorporates only a small amount of learnable parameters for shape generation in M.  Note that the parameters of the SVR model are not tuned for the specific text-to-shape generation task. Therefore, they are not counted in the total number of parameters. Please refer to the below paragraphs (especially the last two paragraphs of this answer)  for more details about this point.
>
> We provide the comparison of learnable model parameters responsible for shape generation and compare it with existing methods as below. We also report the FID and consistency score of compared methods.
>
> |      |    Number of parameters  (M)    | FID(↓) | FPD(↓) | Consistency Score(↑)
> | ----------- | --------------------- | ---- | ---- |------- |
> | CLIP-Forge | Normalized flow network: 18.37  |162.87|37.43| 41.83±17.62|
> |Dream Fields |0.61|181.25|N.A.|25.38±12.33|
> |Our approach|Mapper: 2.43|**124.42±5.11**|35.67±1.09|60.0±10.94|
> |Our with a lightweight mapper|**Mapper: 0.46**|129.01|**34.26**|**67.21±10.64**|
>
> The learnable parameter of CLIP-forge (Sanghi et al., 2022) is **8** times larger than ours.  Note that we do not include the parameters of the CLIP-Forge auto-encoder for a fair comparison. With much fewer parameters, our model outperforms CLIP-Forge in all evaluated metrics. This demonstrates that our performance gain is not purely from the learnable parameters.
>
> To better compare with Dream Fields (**0.61** M parameters), we design a lightweight mapper to match the total number of parameters. Specifically, the mapper is composed of three fully connected layers with **512**, **256**, and **256** output dimensions, yielding a total of **0.46** M parameters which is smaller than Dream Field. With this new mapper, we still outperform Dream Fields in all evaluated metrics. Please see Figure 16 in the supplementary material for quantitative comparison. This further demonstrates our major performance gain is not from the learnable parameters.
>
> From the experiment, we can consistently observe that our model can achieve much better performance with much fewer parameters, manifesting the efficiency of our proposed image as stepping-stone pipeline that allows us to leverage the 3D priors in pre-trained SVR models to enable text to 3D shape synthesis without requiring paired text and 3D data.
>
> Please note that we exclude the number of parameters of the decoder D in our comparison because the only effect of fine-tuning D is to make the background white and does not contribute to the generative capability of our model. As shown in Figure 2 of the main paper, mapper M and decoder D are trained with their own losses **separately** at the same time by **stopping the gradients** from L_D and L_{bg} to propagate to M. Also, to show that fine-tuning D does not improve its generative capability, we feed the same input feature f_s to D before and after fine-tuning, they generate almost the same 3D shape as the original SVR model, as shown in Figure 17 in the supplementary material. And in stage 2, D is not optimized. Admittedly, our generative capability benefits from the SVR model, including D. However, D is not optimized for our text-to-shape generation task. Therefore, we exclude the number of parameters of decoder D for a fair comparison.
>
> We also include the above analysis in our supplementary file Section 9.

---

### Official Review · Reviewer_HSBz · 2022-11-01

**Confidence:** 5
**Correctness:** 3
**Technical Novelty And Significance:** 2
**Empirical Novelty And Significance:** 3
**Recommendation:** 6

**Clarity, Quality, Novelty And Reproducibility:**

Clarity:
+ Is the image encoder Es fixed or optimized during step 1? In Figure 2, it is better to provide the "lock/unlock" icon to Es, Ei, and ET as well, to make the notations consistent.

+ Is the vector direction in Figure 3 (c) reasonable? In step 2, the 𝑀(𝑓𝑇) is optimized to 𝑀(𝑓𝐼), but the vector direction is moving to some other point in the latent space.

+ Analyse and Evaluation: How many shapes are generated from the text prompt at the evaluation step? I'm also concerned with the number of instances used for evaluation. If only four texts per ShapeNet category are used, there will be a total of 52 shapes, which is a very limited number of instances to calculate FPD and FID. I suggest the author run the metric multiple times and provide a mean and a derivation.

+ In the paper, the author mentioned referring to Figure 2 (c)  & Figure 8 (b) for some results, but there is no  Figure 2 (c) or Figure 8 (b) in the paper. Is that false referred?

+ Ablations of background loss. Is the background loss removed at stage 1, stage 2, or both?

Quality
+ Fair. But the paper is hard to follow for me because of lacking background, unexplained symbols and notations, and false references.

Novelty.
+ Good.

Reproducibility.
+ Good.

**Strength And Weaknesses:**

Strength:

+ The paper proposes a solution to using images as a stepping method for text2shape generation.

+ As a zero-shot generation method, it can generate shapes of diverse categories relatively quickly (85 seconds).

+ Using latent space makes the generation and stylization step very flexible.

Weakness:

- The paper focuses on 'latent space mapping'. However, it didn't provide a systematic evaluation/visualization of how the latent space is mapped. Figure 3 (c) provides only one instance which is not convincing enough for me.


- The paper is hard to follow for me because of the writing. For example: (1) In equation 3, the symbol p jumped out without explanation. To make the paper self-contained, it's better to provide an explanation for all the symbols in the equations. (2) The paper lacks enough background knowledge about the SVR model, like the rendering method and reconstruction losses.

**Summary Of The Paper:**

The paper works on text2shape generation. It proposes to use images as a 'stepping stone' of the generation process. The authors use two mapping processes to map image-shape latent space and image-text latent space sequentially. Furthermore, the authors show the model can stylize both texture and shape structure. The experiment shows the method outperforms two zero-shot text2shape generation baselines.

**Summary Of The Review:**

The paper proposes an interesting solution for the text2shape generation problem. I like the method itself. However, the author could better explain the background, notation, and equations better. I'm willing to improve the score if the author could clarify my concerns.

---

> ### Author Response · Authors · 2022-11-18
> **Feedback to Reviewer HSBz: Part 3**
>
> **Q3**: Clarity
>
> **Q3-1**: Is the image encoder Es fixed or optimized during step 1? In Figure 2, it is better to provide the "lock/unlock" icon to Es, Ei, and ET as well, to make the notations consistent.
>
> **A3-1**: Thanks for your suggestion. All the Es, Ei, and ET are locked in all stages in Figure 2. We have revised Figure 2 to improve its clarity and provide more description in the figure caption.
>
> **Q3-2**: Is the vector direction in Figure 3 (c) reasonable? In step 2, the 𝑀(𝑓_𝑇) is optimized to 𝑀(𝑓_𝐼), but the vector direction is moving to some other point in the latent space.
>
> **A3-2**: Thanks for your comment. We make this figure plot just for visualization.  The visualized direction may not truly reveal the feature mapping direction in the high-dimensional space. To make it clear, we have redrawn Figure 3(c) to avoid confusion. Please see the new Figure 3(c) in the revised version.
>
> In addition, as shown in Answer to Q1 above of this rebuttal, our stage 2 alignment can significantly reduce the difference between the mapped text and the GT shape feature from **0.45** to **0.17** on average, further demonstrating the effectiveness of our stage-2 alignment on feature space mapping.
>
> **Q3-3**: Analyze and Evaluation: How many shapes are generated from the text prompt at the evaluation step? I'm also concerned with the number of instances used for evaluation. If only four texts per ShapeNet category are used, there will be a total of **52** shapes, which is a very limited number of instances to calculate FPD and FID. I suggest the author run the metric multiple times and provide a mean and a derivation.
>
> **A3-3**: Thanks for your suggestion. We repeat our experiments additionally two more times. The FIDs that we obtained are **127.28**, **128.68** and **117.29**, and FPD are **36.03**, **36.74** and **34.26**. The results (FID **124.42 ± 5.11** and FPD **35.67 ± 1.09**) are consistent with our presented numbers (FID **128.68** and FPD **36.03**). We have included the results in Table 1 of the main paper.
>
> **Q3-4**: In the paper, the author mentioned referring to Figure 2 (c) & Figure 8 (b) for some results, but there is no Figure 2 (c) or Figure 8 (b) in the paper. Is that false referred?
>
> **A3-4**: For Figure 2, please note that the notations (a,b,c) are on the left, instead of the bottom. Please refer to Figure 2(c) S on the right-hand-side of the figure from the text “A mahogany table”.
>
> For Figure 8(b), we are sorry for the mistake in the reference. It should be Figure 9.
>
> **Q3-5**: Ablations of background loss. Is the background loss removed at stage 1, stage 2, or both?
>
> **A3-5**: In the ablation study, the background loss is removed in both stage-1 and stage-2 feature space alignment.
>
> Here, we added additional experiments to show the results for removing L_{bg} in stage 1 and stage 2 respectively.  The results in Figure 1 of the main paper show that the background loss in both stage 1 and stage 2 is necessary for our approach.  Also, even though stage-1 alignment has encouraged the background to be white, we still need this loss in stage 2 to obtain satisfying results. We have included the results in Table 1 of the main paper.
>
> | Method| FID (↓)| Consistency Score) (↑)| FPD (↓)|
> | ---------- | ---- |----- |--- |
> | w/o L_{bg_1}|  149.45| 29.45 ± 14.67| 40.85|
> |w/o L_{bg_2}|156.52|31.55 ± 8.87|38.31|
> |w/o L_{bg}|178.34|30.96 ± 15.49|40.98|

---

> ### Author Response · Authors · 2022-11-18
> **Feedback to Reviewer HSBz: Part 2**
>
> **Q2-1**: In equation 3, the symbol p jumped out without explanation.
>
> **A2-1**: It seems there is no symbol p in Equation 3.
>
> If you mean Equation 2, the symbol p in this equation means the query point. Please see the two lines below Equation 2 in both of our original and revised submissions: “p is a query point; ray(o, p) ∩ F = ∅ mean … ”.
>
> Besides, we have summarized all the symbols and notations in Table 5 of the supplementary material.
>
> **Q2-2**: The paper lacks enough background knowledge about the SVR model, like the rendering method and reconstruction losses.
>
> **A2-2**: Thanks for your comment. We have added a review of SVR models and rendering methods in the supplementary material Section 8.
>
> **SVR model:**
>
> Single-view reconstruction (SVR)  aims to reconstruct the 3D shape from a single-view image of it. Recently, many approaches have been proposed for meshes [1,2,3], voxels [4,5], and implicit 3D shapes [6,7]. Recently, to extend SVR to in-the-wild categories, Alwala et al. [8] proposed a new approach called SS3D to learn 3D shape reconstruction using single-view images for the reconstruction of hundreds of categories.
>
> As 3D-to-2D projections, single-view 2D images are more closely related to shapes than texts, since they reveal many attributes of 3D shapes, e.g., structure, details, appearance, etc. The strong correlation between 2D images and 3D shapes motivates us to reduce the challenging text-to-shape generation task to text-to-image and then SVR, by connecting the CLIP features from text with the shape features in SVR using images as an intermediate step to gradually bridge the gap between text and shape.
> Specifically, we extend the pre-trained SVR model to be compatible with text input, transforming the challenging text-to-shape task into SVR. Our framework can work with different SVR approaches to extend them for 3D shape generation from texts. So, our approach is orthogonal to the SVR approaches.
>
>
> **Differentiable Rendering:**
>
> 3D rendering is an important topic in computer vision and graphics. It takes a 3D scene as input and predicts a 2D image of a given camera pose. Beyond 3D rendering, differentiable rendering further aims to derive the differentiations of the rendering function. With differentiable rendering, a renderer can be integrated into an optimization framework, thus a 3D shape can be reconstructed from multi-view 2D images. Neural Volume Rendering [9] and its following works [10,11] aim to synthesize novel view images of a 3D scene.
>
> Besides, recent works [12,13] leverage differentiable rendering for 3D shape generation using 2D images. In this work, we derive the 2D images of the generated 3D shape using differentiable rendering and use a pre-trained large-scale image-language model CLIP to encourage the 2D images to be consistent with the input text. Thanks to differentiable rendering, we can update the generated 3D shape indirectly using rendered images.
>
> [1] N. Agarwal and M. Gopi. Gamesh: Guided and augmented meshing fordeep point networks. In 3DV, 2020.
>
> [2] W. Chen, H. Ling, J. Gao, E. Smith, J. Lehtinen, A. Jacobson, and S. Fidler. Learning to predict 3D objects with an interpolation-based differentiable renderer. NeurIPS, 2019.
>
> [3] N. Wang, Y. Zhang, Z. Li, Y. Fu, W. Liu, and Y.-G. Jiang. Pixel2mesh: Generating 3D mesh models from single rgb images. In ECCV, 2018.
>
> [4] H. Xie, H. Yao, X. Sun, S. Zhou, and S. Zhang. Pix2vox: Context-aware 3D reconstruction from single and multi-view images. In ICCV, 2019.
>
> [5] N. Zubic and P. Liò. An effective loss function for generating 3D models from single 2D image without rendering. arXiv preprint arXiv:2103.03390, 2021.
>
> [6] Z. Chen and H. Zhang. Learning implicit fields for generative shape modeling. In CVPR, 2019
>
> [7] M. Li and H. Zhang. D^2IM-Net: Learning detail disentangled implicit fields from single images.CVPR, 2021.
>
> [8] K. V. Alwala, A. Gupta, and S. Tulsiani. Pre-train, self-train, distill: A simple recipe for supersizing 3d reconstruction. CVPR, 2022.
>
> [9] Mildenhall, Ben, et al. "Nerf: Representing scenes as neural radiance fields for view synthesis." ECCV, 2020.
>
> [10] Barron, Jonathan T., et al. "Mip-nerf: A multiscale representation for anti-aliasing neural radiance fields." ICCV, 2021.
>
> [11] Jain, Ajay, Matthew Tancik, and Pieter Abbeel. "Putting nerf on a diet: Semantically consistent few-shot view synthesis."
> ICCV, 2021.
>
> [12] Niemeyer, Michael, et al. "Differentiable volumetric rendering: Learning implicit 3d representations without 3d supervision." CVPR, 2020.
>
> [13] Gao, Jun, et al. "GET3D: A Generative Model of High Quality 3D Textured Shapes Learned from Images." NeurIPS, 2022.

---

> ### Author Response · Authors · 2022-11-18
> **Feedback to Reviewer HSBz: Part 1**
>
> **Q1**. The paper focuses on 'latent space mapping'. However, it didn't provide a systematic evaluation/visualization of how the latent space is mapped. Figure 3 (c) provides only one instance which is not convincing enough for me.
>
> **A1**. Thanks for your insightful comments. To provide more insights on explaining how the latent space is mapped, we measure the distance of features at different stages on all the samples in our test set on ShapeNet. The notations follow Figure 3 (c) of the main paper, M means the mapper, and d means cosine distance.  Our ultimate goal is to obtain a text mapper M’ (Figure 2 in the main paper) to map the text feature space f_T to shape feature space f_S. Our key insight is to use the image with features f_I as a stepping stone to gradually narrow their distances using the two-stage mapping. Note that the image f_I and text features f_T are obtained using pre-trained CLIP models.
>
>
> |                       | d(M(f_I),M(f_T))  | d(M(f_I), f_S)) | d(M(f_T), f_S)) | d(M’(f_T), f_S)) |
> |-------------------|-----------|-----------|------------|------------|
> |Distance (mean ± std) |   0.58 ± 0.23   |  0.21 ±  0.10   |  0.45  ±  0.20   |   0.17 ±  0.08   |
>
> In the stage-1 alignment process, we train a mapper M to map image features f_I to a space M(f_I) close to the shape space f_S using image data and the regression loss L_M. Note that the text feature f_T and image feature f_I are all from the CLIP model in a shared embedding space. It’s natural that the trained mapper can be used to map the text feature f_T to M(f_T), making text features closer to the shape space. However, when measuring the distance among M(f_I),  M(f_T), we find the average distance among all samples is d(M(f_I), M(f_T)):  **0.58 ± 0.23**,  the average distance between M(f_I)  and f_S is d(M(f_I), f_S)): **0.21 ± 0.10**, and the average distance between shape and text is d(M(f_T), f_S))： **0.45  ±  0.20**. This implies that there is a gap between the CLIP image and text feature after the first step mapper and further motivates our stage-2 alignment. Note that there is no GT shape for our task on 3D generation, so we manually select a shape in the ShapeNet dataset that matches the input text as the GT.
>
> In the stage-2 alignment, M is further updated and the final delivered mapper is called M’ which is to further narrow down the gap between mapped text features and shape features. The average distance between the mapped text feature M’(f_T) and shape feature space f_S: d(M’(f_T), f_S)): **0.17 ±  0.08** which is much smaller than the corresponding distance after stage-1 alignment.  It shows that stage 2 alignment can significantly reduce the difference between the mapped text and the GT shape feature from **0.45** to **0.17** on average.
>
> We also provide this in the supplementary material. Please refer to Section 7 and Table 3 in the supplementary material for more details on how features are mapped for each sample.

---

### Author Response · Authors · 2022-11-18
**Author Feedback to All Reviewers of Paper 420**

We thank all reviewers for the thoughtful suggestions. We found most questions and concerns are on the unclear presentation and missing details. In this rebuttal, we will provide clarifications and details, and we will strive to revise the paper to **provide details** and to **improve its clarity**.

To start, we want to first stress that

(1) We will *RELEASE CODE* and the trained models on GitHub upon the publication of this work to facilitate future research.

(2) This work tackles the *VERY CHALLENGING* task of generating 3D shapes from text WITHOUT paired text and 3D data. This is an *IMPORTANT* research direction, since doing so takes out the need of preparing large paired text and 3D shape collections for training the network model.

(3) Our network is lightweight and efficient, as well as flexible to work with different SVR models. Also, it has GOOD PERFORMANCE, BOTH quantitatively and qualitatively, over the two most recent works CLIP-Forge [32] and Dream Fields [11].  Many of our presented results *CANNOT BE ACHIEVED* by the existing works.

(4)  We followed the suggestions of reviewers Z3WY and CfJ1 to adopt our model to work with the very recent approach GET3D. Results in Figure 11 of the main paper and Figure 11 of the supplementary material show that our method can ALSO WORK effectively with GET3D to produce high-quality 3D shapes from the text.

We have taken your suggestions and made the following changes to our previous draft, with the main changes marked in blue.

In the main paper:

(1) Added high-quality text-guided 3D shape generative results with GET3D in Figure 11.

(2) Revised Figure 2 and added some necessary details to improve the clarity.

(3) Revised Figure 3 (c) for better visualization and use the average distance of all samples in the test set.

(4) Table 1: change “Category” to “Method type”.

(5) Added ablation studies for removing the background loss in Stage 1 and Stage 2 respectively.

(6) Repeated our experiments three times and reported the mean values and standard deviations.

(7) Added two realistic stylization results in Figure 9.

(8) Fixed the wrong figure reference Figure 8 (b) to be Figure 9.

In the supplementary material:

(1) Added implementation details on camera pose setting and decoder duplication in Section 2.

(2) Added details on FPD and FID metrics in Section 2.

(3) Added additional generative results built upon GET3D in Figure 11.

(4) Added an analysis on feature space mapping, and list how feature distances change in two-stage feature space alignment for all samples in the test set in Section 7.

(5) Reviewed related works on single-view reconstruction and differentiable rendering in Section 8.

(6) Added an analysis on the number of parameters and added some details on model training in Section 9.

(7) Summarized symbols and notations used in the paper in Section 13.

---

### Decision · Program_Chairs · 2023-01-20

**Decision:**

Accept: notable-top-25%

**Justification For Why Not Higher Score:**

Reviewers initially had concerns about some aspects of the work, but increased their ratings to weak accept after the rebuttal.  The method is interesting but is fairly intuitive combination of advances in joint text-image embeddings and image-to-3D.

**Justification For Why Not Lower Score:**

The proposed method is flexible and will encourage additional work in the area of text-to-shape generation.

**Metareview: Summary, Strengths And Weaknesses:**


Summary:
The paper presents a framework for generating 3D shapes from text by utilizing: 1) a pretrained single-view image to 3D shape reconstruction (SVR) model 2) pretrained model that gives a joint embedding of image-text (e.g. CLIP), and training a feature alignment network to map the CLIP features to the feature space of the SVR model.  The authors also propose an optional text-guided stylization step that furthur enriches the texture of the shape.  Experiments show that the proposed method can generate reasonable shapes for an given input text.

Strengths:
- The submission provides a flexible framework for utilizing pretrained vision-language language + SVR models to generate 3D shapes without paired text-to-3D data
- The proposed method is fast compared to prior work such as DreamFields that requires per-prompt optimization
- The paper can stimulate additional work in the field of text-to-3D shape generation which is in its nascent stages,

Weaknesses:
- The quality of the generated shapes are very dependent on the SVR used, with the generated shapes in the initial version very smoothed out and lacking fine detail
- The set of prompts used to generate the shapes are fairly simplistic
- The writing can be improved as some reviewers found the parts of the paper difficult to follow initially, and some parts to be unclear



**Note From Pc:**

if the above contains the word "oral" or "spotlight" please see: "oral" presentation means -> notable-top-5% and "spotlight" means -> notable-top-25%. As stated in our emails, we are disassociating presentation type from AC recommendations